# Application of Raney Al-Ni Alloy for Simple Hydrodehalogenation of Diclofenac and Other Halogenated Biocidal Contaminants in Alkaline Aqueous Solution under Ambient Conditions

**DOI:** 10.3390/ma15113939

**Published:** 2022-05-31

**Authors:** Helena Bendová, Barbora Kamenická, Tomáš Weidlich, Ludvík Beneš, Milan Vlček, Petr Lacina, Petr Švec

**Affiliations:** 1Chemical Technology Group, Institute of Environmental and Chemical Engineering, Faculty of Chemical Technology, University of Pardubice, Studentská 573, 532 10 Pardubice, Czech Republic; helena.bendova@upce.cz (H.B.); barbora.kamenicka@student.upce.cz (B.K.); 2Joint Laboratory of Solid State Chemistry, Faculty of Chemical Technology, University of Pardubice, Studentská 573, 532 10 Pardubice, Czech Republic; ludvik.benes@upce.cz (L.B.); milan.vlcek@upce.cz (M.V.); 3GEOtest, a.s., Šmahova 1244/112, 627 00 Brno, Czech Republic; lacina@geotest.cz; 4Department of General and Inorganic Chemistry, Faculty of Chemical Technology, University of Pardubice, Studentská 573, 532 10 Pardubice, Czech Republic; petr.svec2@upce.cz

**Keywords:** nickel alloy, reductive dechlorination, hydrometallurgy, NaBH_4_, biocide, drug, water treatment

## Abstract

Raney Al-Ni contains 62% of Ni_2_Al_3_ and 38% NiAl_3_ crystalline phases. Its applicability has been studied within an effective hydrodehalogenation of hardly biodegradable anti-inflammatory drug diclofenac in model aqueous concentrates and, subsequently, even in real hospital wastewater with the aim of transforming them into easily biodegradable products. In model aqueous solution, complete hydrodechlorination of 2 mM aqueous diclofenac solution (0.59 g L^−1^) yielding the 2-anilinophenylacetate was achieved in less than 50 min at room temperature and ambient pressure using only 9.7 g L^−1^ of KOH and 1.65 g L^−1^ of Raney Al-Ni alloy. The dissolving of Al during the hydrodehalogenation process is accompanied by complete consumption of NiAl_3_ crystalline phase and partial depletion of Ni_2_Al_3_. A comparison of the hydrodehalogenation ability of a mixture of diclofenac and other widely used halogenated aromatic or heterocyclic biocides in model aqueous solution using Al-Ni was performed to verify the high hydrodehalogenation activity for each of the used halogenated contaminants. Remarkably, the robustness of Al-Ni-based hydrodehalogenation was demonstrated even for the removal of non-biodegradable diclofenac in real hospital wastewater with high chloride and nitrate content. After removal of the insoluble part of the Al-Ni for subsequent hydrometallurgical recycling, the low quantity of residual Ni was removed together with insoluble Al(OH)_3_ obtained after neutralization of aqueous filtrate by filtration.

## 1. Introduction

The presence of biocidal remedies and other fine chemicals in the water sources and the craving for cost-effective techniques applicable for the decomposition of these non-biodegradable pharmaceuticals has become of major interest [1,2]. Concretely, diclofenac (DCF), fluconazole, hydrochlorothiazide, chlorohexidine, ketoconazole, triclosan, atrazine, simazine and a wide range of other halogenated fine chemicals have attracted attention in recent years since they are often being detected in wastewater treatment plants effluents and/or groundwater. These pollutants are only poorly degraded by traditional wastewater treatments due to their biocidal properties caused by halogens bound in their structures [3,4]. The above-mentioned pollutants emerging from the effluents reach the receiving watercourses, and thus they have negative effects on the aquatic fauna and flora.

Chronic toxicity trials performed on the rainbow trout (*Oncorhynchus mykiss*) revealed cytological changes in the liver, kidneys and gills after 28 days of exposure to just 1 µg L^−1^ of DCF. For a concentration of 5 µg L^−1^, renal lesions were evident, as well as drug bioaccumulation in the liver, kidneys, gills and muscles [5,6]. The occurrence of DCF in different aquatic compartments and effluents of wastewater, surface water and groundwater is detected in a wide low-concentration range from 0.8 ng L^−1^ up to 4.4 mg L^−1^ [7,8], which highlighted that the mentioned negative effects are of sufficient magnitude to suspect chronic toxicity in aquatic organisms. Based on such facts, the removal of non-biodegradable, biologically active micropollutants from wastewaters is important to minimize their potential negative effect on all living organisms.

By using separation techniques such as electro-reversible adsorption [9], reverse osmosis [10] or nanofiltration [3,11], the concentration of DCF reaches even hundreds of mg L^−1^ in the aqueous concentrate (or retentate) produced by these separation methods as waste streams [3,9,10,11]. These low volumes of concentrate (retentate), however, should be effectively treated before discharging to the environment. Due to such reasons, various treatment methods enabling the removal of non-biodegradable organic contaminants were proposed earlier for the removal of DCF, including advanced oxidation processes (AOPs) [1,10,12,13,14], electron beam irradiation [15] and/or reduction (hydrodehalogenation (HDH)) processes [4,10].

Generally, the main disadvantage of oxidation processes involves the consumption of high excess oxidants for the formation of biodegradable products. In addition, many of the published papers dealing with the application of AOPs for DCF removal only quantified DCF content in the aqueous solution obtained after tested AOP method treatment [16,17,18,19] (Table 1). Moreover, some papers mentioned the high toxicity of formed primary oxidation or photolysis products [20,21,22].

HDH seems to be, in contrast, a more environmentally friendly process, which allows a significant reduction in the toxicity of the contaminated water and avoids the formation of even more harmful intermediates. The formation of appropriate hydrocarbon from the halogenated contaminant allowed for a dramatic decrease in the ecotoxicity of the aqueous solution obtained after dehalogenation and neutralization to negligible values and enabled their subsequent biodegradation [4,15,23,24].

In most cases, very poison-sensitive platinum-based catalysts are described for the HDH as very expensive species that are, unfortunately, prone to catalyst fouling or poisoning [4,25]. As Nieto-Sandoval et al. observed, even a higher concentration of inorganic chlorides caused the deactivation of used palladium catalysts, which further disadvantages these expensive catalysts for the real applications in wastewater treatment techniques [4].

In addition, noble metals and their soluble compounds proved to be very toxic to aquatic fauna [26,27,28].

The aim of this work was to demonstrate the possibilities of the utilization of the commercial Raney Al-Ni alloy for HDH of the persistent DCF and other halogenated biocides in model and even in real wastewater as the suitable process for the formation of dechlorinated biodegradable products [3,9,10,11]. Commercially available Raney Al-Ni alloy was used because of its observed high HDH activity, which is comparable with platinum- or palladium-based catalysts [4,29,30,31,32]. Furthermore, we proved earlier that the HDH activity of Raney Al-Ni is higher when compared to Al-Cu alloys such as Devarda’s alloy [33]. In an alkaline aqueous solution, Al works as the reductant (i.e., hydrogen generator) and nickel as the HDH catalyst.

After the HDH process, subsequent neutralization of obtained aqueous phase is accompanied by coagulation and flocculation of dissolved Al(III) species, which remove even low concentrations of Ni particles from the resultant reaction mixture (i.e., adsorption of Ni metal particles onto precipitated Al(OH)_3_). In comparison with platinum-based metals, nickel is described as much less toxic [34,35], and as was described earlier by our group [31], its concentration after the neutralization step, accompanied by removal of precipitated aluminum hydroxide by filtration, is below 10 µg Ni L^−1^. In the term of this work, a complete operating condition study was carried out, and a kinetic model was accordingly proposed. The recyclability of the used reductant was also evaluated. The versatility of the studied system was demonstrated using a mixture of nine different halogenated biocides dissolved in an aqueous solution and real environmentally relevant aqueous matrices, such as hospital wastewater spiked with DCF.

## 2. Materials and Methods

### 2.1. Materials

Diclofenac sodium salt (NaDCF, 98%+), fluconazole (98%+), hydrochlorothiazide (98%+), chlorhexidine (99%+), ketoconazole (98%+), triclosan (Irgasan, 97%+), atrazine (analytical standard), aluminum–nickel alloy (purum, 50% Al basis, 50% Ni basis) and an aqueous solution of NaBH_4_ (12 wt.% in 14 M NaOH) were delivered by Merck Co. (Prague, Czech Republic). Deuterated chloroform (CDCl3) was purchased from Merck Co. (Prague, Czech Republic). Additional chemicals and solvents in p.a. quality were obtained from a local supplier (Lach-Ner Co., Neratovice, Czech Republic).

A sample of hospital wastewater was stabilized with HNO_3_ for AOX (adsorbable organically bound halogens) determination [36]. The parameters of hospital wastewater stabilized with HNO_3_ are: pH = 2.07; [NH_4_^+^] = 32.4 mg L^−1^; [Cl^−^] = 820.7 mg L^−1^; [Ca^+2^ + Mg^+2^] = 4.5 mM; [NO_3_^−^] = 1213.6 mg L^−1^; COD_Cr_ = 349 mg O_2_ L^−1^; AOX = 1.44 mg Cl L^−1^. The experiments were carried out using deionized water, except where noted otherwise. All operations and manipulations were conducted in the air.

### 2.2. Experimental Procedure

Laboratory HDH trials were carried out in a magnetically stirred 250 mL round bottom flask at 25 °C. During the reaction, the flask was immersed in a thermostated water bath placed on a magnetic stirrer Heidolph HeiStandart with temperature sensor Pt1000. The 4 mM aqueous NaDCF solution was mixed with an appropriate amount of 0.5 M aq. KOH and filled with water to obtain a 200 mL reaction mixture in which the initial NaDCF concentration was 2 mM (0.59 g L^−1^), and an appropriate amount of reductant was added (see Table 2 and Figure 1b). The stirring velocity was set to 750 rpm. After the overnight stirring, the residual undissolved part (Ni slurry) was removed from the reaction mixture by simple filtration. The filtrate was subsequently acidified by the addition of 16 wt.% H_2_SO_4_ to pH = 2–3 and analyzed by HPLC. The accurate quantity (50 mL) of filtrate of the reaction mixture was extracted by dichloromethane (3 × 20 mL), and the organic phase was evaporated to dryness and dissolved in an appropriate solvent for subsequent NMR analysis (CDCl_3_).

The comparative experiments and kinetic studies were performed in 250 mL round-bottomed flasks equipped with magnetic stirring on Starfish equipment (Radleys Discovery Technologies, Saffron Walden, UK) installed on a magnetic stirrer Heidolph Heistandard for parallel reactions. The reaction flasks were closed by a tube filled with granulated charcoal.

### 2.3. Analytical Methods

The content of DCF and its reduction products was determined by HPLC (Agilent 1260 Infinity II, Nucleosil C18 HPLC 5 μm 250 × 3.2 mm Column, DAD detector, Poway, CA, USA). An amount of 5 mL of liquid samples was periodically withdrawn from the reaction vessel, and after acidification, with 1 M HCl to pH = 8–10 (to prevent the precipitation of Al(OH)_3_), dilution to 25 mL volume with water and subsequent filtration of nickel slurry (0.45 μm membrane filter), the samples were analyzed. The analyses were carried out at 283 nm using 3/7 (*v*/*v*) acetonitrile/water as the mobile phase (flow rate 0.9 mL min^−1^). Moreover, the conversion of DCF to APA was monitored by multinuclear NMR spectroscopy. The NMR spectra were recorded from solutions in CDCl_3_ and DMSO-d_6_ on a Bruker Ascend^TM^ 500 spectrometer (equipped with *Z*-gradient 5 mm TBI 500 MHz S1 probe, Karlsruhe, Germany) at frequencies 500.13 MHz for ^1^H and 125.76 MHz for ^13^C{^1^H} at 295 K (see Appendix A for details).

The content of Al and Ni in the treated water solutions was determined with a GBC Integra XL ICP OES spectrometer iCAP 7400 D (Thermo Scientific, Dreieich, Germany). The BOD_5_ (biological oxygen demand) and COD_Cr_ (chemical oxygen demand) values of the obtained filtrates were determined using Hach–Lange cuvette tests (LCK 555 for BOD_5_ determination), LCK 514 (for COD_Cr_ determination). AOX analyses were performed according to ISO 9562 standard (European ISO 9562, 2004).

The images of samples were obtained using scanning electron microscopes JEOL JSM-5500LV and JEOL JSM7500F (Akishima, Tokyo, Japan).

## 3. Results and Discussion

### 3.1. HDH of DCF Using Raney Al-Ni Alloy

In the preliminary set of experiments, optimization of the ratio of the reagents was determined. For the hydrodechlorination (HDC), KOH was used instead of NaOH as the source of biogenic potassium necessary for subsequent bio-treatment of the target HDC product. The results are presented in Table 2. In our work, similarly to Pd^0^-based HDC [4], (2-(2-chloroanilino)-phenylacetate (Cl-APA) was the sole HDC intermediate formed by substitution of one of the chlorine bound within the DCF structure by hydrogen. In the subsequent HDC step, Cl-APA was further reduced to provide 2-anilinophenylacetate (APA) (Figure 1).

Differences in HDC efficiency of DCF using various reaction conditions were determined (Table 2). Furthermore, the comparison of DCF reductive removal is depicted in Figure 1a–d (and in the Appendix A). Based on results obtained within run 14 in Table 2, the application of powdered aluminum without Ni is not effective for HDC of DCF. This observation supports that nickel poses a key role in this described HDC process.

The minimal excess of reactants for complete HDC of DCF after 16 h of vigorous stirring at room temperature is at least 10 mol of Al used in the form of Al-Ni alloy and 25 mol of KOH per 1 mol of DCF dissolved in water (in 2 mM aq. DCF) (see Table 2, run 10). By using tap water, 1 mol of DCF was completely reduced to APA using 15 mol of Al (used as Al-Ni alloy) together with 75 mol of KOH (see run No. 3 in Table 2). As can be seen in Figure 1, by using 15 mol of Al in Al-Ni alloy and 75 mol of KOH per 1 mol of DCF, the full HDC was completed after 40 min of producing APA as the final organic sole product (Table 2, run No. 1 and Figure 1). At lower quantities of Al-Ni and KOH, the reaction rate is much slower.

The conversion of HDC reaction (removal of AOX) is strongly dependent on the quantity of produced soluble KAl(OH)_4_ analyzed by ICP-OES as dissolved Al (see Appendix A). Practically, at least 2.6 mmol of Al (68 mg, 340 mg L^−1^) should be dissolved by a reaction of Al-Ni with KOH and DCF to accomplish a complete HDC of 0.4 mmol of DCF under efficient stirring. It means that complete HDC of 1 mol DCF is accompanied by dissolution of 6.5 mol Al from a total of 10 mol Al added in the form of Al-Ni alloy (Table 2, run No. 10).

Possible mechanisms of the described HDH comprise either (a) in situ generations of hydrogen by the dissolution of Al from Ni_2_Al_3_ or 38% NiAl_3_ crystalline phases occurring in Al-Ni alloy and formation of the catalytically active Raney nickel saturated with produced H_2_ as the active HDH agent or (b) direct HDH of DCF adsorbed on Al-Ni alloy acting as a galvanic couple, which hydrodehalogenates DCF during oxidation and dissolution of Al. The dissolution of Al enables denudation of the active Al-Ni surface for further HDH process [30].

We observed earlier [27] that the HDC mechanism (HDC based on in situ produced Raney Ni) works in the case of HDC of several chlorinated anilines. In this case, the positive role of glucose as an additive was documented, which influences the formation of Raney Ni nanoparticles from used Al-Ni and enables effective HDH. For verification of this reaction pathway, glucose was added in different quantities to the mixture of DCF dissolved in diluted KOH solution before the addition of Raney Al-Ni. Unfortunately, however, for DCF, we observed that the addition of glucose to the reaction mixture only causes a decrease in the HDC rate (see Figure 1c). The attempts to decrease the required quantity of Al-Ni by the addition of glucose acting as a promoter of highly reactive Ni^0^ nanoparticles formation (as we published earlier [37] were not successful. This fact could indicate the possible HDC mechanism of DCF, which is based not on a reaction with the activated hydrogen adsorbed on (nano)nickel surface, as was assumed for HDC of 3-chloroaniline [32], but was in all probability caused by the nickel catalyzed reductive action of aluminum in alkali after adsorption of DCF on the surface of the Al-Ni alloy composing local galvanic couples between active Al anode and Ni cathode (Ni catalyzed dissolving-metal reduction) [30].

To corroborate the hypothesis of the mechanism described, vide supra, the efficiency of adsorption of DCF on the Al-Ni surface was proved using a neutral 2 mM aqueous solution of NaDCF (0.635 g L^−1^) and 1.65 g L^−1^ of Al-Ni alloy in the absence of KOH (see Appendix A for the experimental data). It was found that the concentration of DCF along the adsorption experiment had decreased by ca. 25% after 30 min. These results thus corroborated the contribution of DCF adsorption on the Al-Ni surface to cause a subsequent HDC reaction on the Al-Ni surface according to the proposed mechanism.

This phenomenon could be explained by the different course of corrosion of the Al-Ni alloy with alkali metal hydroxides after the addition of glucose to the alkaline aqueous reaction mixture. The action of 1 wt.% aqueous KOH solution causes deep (pitting) corrosion of Al-Ni, while the addition of glucose to these mentioned alkali metal hydroxide solutions only induces superficial corrosion of Al-Ni, which is accompanied by the slowdown of the dissolution of Al from the Al-Ni alloy and the subsequent decrease in the HDC reaction rate (see Figure 2a–c and Appendix A for the experimental data).

### 3.2. Kinetic Modelling

For a comparison of reaction rates, a simple kinetic model based on the reaction pathway shown in Figure 1, considering the pseudo-first-order, was employed.

The following kinetics equations were proposed
(1)$$r_{DCF}=\frac{-dc_{DCF}}{dt}=k_{1}c_{DCF}$$
(2)$$r_{Cl-APA}=\frac{dc_{Cl-APA}}{dt}=k_{1}c_{DCF}-k_{2}c_{Cl-APA}$$
(3)$$r_{APA}=\frac{dc_{APA}}{dt}=k_{2}c_{Cl-APA}$$
where *C_DCF_*, *C_Cl-APA_* and *C_APA_* are concentrations of DCF, Cl-APA and APA in the solution and *k*_1_ and *k*_2_ are the apparent first-order rate constants (see Figure 1 in the main document). The integration of the Equations (1)–(3) was performed with the following initial values *t* = 0: *C_DCF_* = *C_DCF_*_,0_; *C_Cl-APA_* = *C_APA_* = 0 and Equations (4)–(6) were obtained
(4)$$c_{DCF}=c_{DCF,0}e^{-k_{1}t}$$
(5)$$c_{Cl-APA}=c_{DCF,0}\frac{k_{1}}{k_{2}-k_{1}}\left(e^{-k_{1}t}-e^{-k_{2}t}\right)$$
(6)$$c_{APA}=\frac{c_{DCF,0}}{k_{2}-k_{1}}\left[k_{2}\left(1-e^{-k_{1}t}\right)-k_{1}\left(1-e^{-k_{2}t}\right)\right]$$

The experimental data were simultaneously fitted to Equations (4)–(6).

The following figures (Figure 1a–d and Appendix A) depict the kinetic studies of selected experiments. One chromatogram from HPLC showing unambiguous separation of DCF, Cl-APA and APA is also included.

The values of kinetic constants *k*_1_ and *k*_2_ were obtained by minimizing the standard deviation of experimental and calculated values of concentrations of all components of the mixture (DCF, Cl-APA and APA). The standard deviation was within limits from 6% to 20%.

The values of the resulting kinetic parameters are summarized in Table 3. By taking into account the data, one can calculate that for a complete HDC of 1 mol of DCF, it is necessary to use 20 mol of Al and 9.34 mol of Ni in Al-Ni alloy and 25 mol of KOH (see results in Table 3). The proposed pseudo-first-order kinetic model is in good agreement with the experimental results, as can be seen from the example depicted in Figure 1a–d for run No. 1 (and in Appendix A).

### 3.3. Applicability of Used Al-Ni Alloy for HDC

An important shortcoming of HDC is related to the deactivation of the reducing agent Al-Ni (compare runs No. 2 and 15 in Table 2), which is caused by the significant loss of Al during the first reduction step. Thus, the activity of the spent Al-Ni alloy was assessed upon sequential re-use after simple separation of the used Al-Ni slurry from the aqueous phase by sedimentation.

In the subsequent reaction of 2 mM aq. DCF in 150 mM KOH with re-used Al-Ni, only incomplete conversion was achieved after 18 h of stirring at 25 °C (the reaction mixture contains 39% DCF, 8% Cl-APA and 53% APA, AOX = 55 mg Cl L^−1^), which corresponds well with the dissolution of 183 mg Al L^−1^ (only 1.35 mmol remaining Al was dissolved in the reaction mixture after 18 h in the second run; see Table 2, run 2).

Due to the low content of the remaining active Al in the once used Al-Ni alloy, the HDC using an excess of external reductant NaBH_4_ (31 mmol) in 0.96 M NaOH was performed (see runs No. 3 and 15 in Table 2). After 18 h of vigorous stirring, 89% APA, 3% Cl-APA and 8% of DCF content were determined by HPLC in the resulting reaction mixture, and the content of AOX value decreased from 123.5 12.5 mg Cl L^−1^ to AOX = 12.5 mg Cl L^−1^. Under the severe conditions used in run No. 15, only microcrystals of cubic Ni (no residual Al) were determined by the X-ray powder diffraction in separated and dewatered Ni slurry Figure 2a–c show the morphology of the fresh and used Al-Ni alloy particles applied in this study without and with the addition of glucose, Figure 2d action of NaBH_4_ in NaOH (used in excess) on used Al-Ni alloy (Run No. 15 in Table 2). (In the Appendix A, the corresponding XRD analyses and Al content of fresh Al-Ni alloy and used Al-Ni are given in Appendix A.) 

These results are not sufficient for exhaustive HDC using recycled Al-Ni alloy, but its use hopefully enables potential pretreatment (partial HDC) before the final complete HDC induced by the subsequent addition of a low quantity of the fresh Al-Ni. In order to recycle deactivated Ni^0^, the hydrometallurgical treatment of spent Ni slurry after the HDC by K_2_S_2_O_8_/H_2_SO_4_ was studied, producing NiSO_4_ [38].

NiSO_4_, obtained from the spent Al-Ni alloy, was successfully tested as a source of active Ni^0^ catalyst produced in situ by Al^0^/KBH_4_ reduction used in excess of subsequent HDC of DCF, but even in this case, the incomplete HDC of DCF was observed [38]. Using a combination of lower quantities of NiSO_4_/NaBH_4_ without the addition of Al only causes, in contrast, low conversion of DCF to APA with high content of Cl-APA (see Table 2, run No. 16).

### 3.4. Applicability of Ni-Based HDC for Treatment of Water Contaminated with Broad Spectrum of Halogenated Biocides

In order to demonstrate the concept of applicability of Al-Ni within the reductive treatment of wastewater contaminated with a broad spectrum of halogenated biocides, HDC of nine different halogenated aromatic and heterocyclic biocides was investigated (Table 4).

Because atrazine and simazine are the common aqueous contaminants, their broad application as cost-effective and highly efficient herbicides are due to their high mobility and persistence in the environment [39]. Gawel et al. published that nanoparticles of iron are not effective for hydrodehalogenation of atrazine, and Fenton oxidation must be used for its removal [39]. Leonard et al. described nickel catalyst effective for HDC of atrazine caused by pressurized gaseous hydrogen at 5 MPa and 140 °C [40]. Both atrazine and simazine are detoxified by nucleophilic substitution of bound chlorine by the action of alkaline hydrogen peroxide solution [41].

Cetirizine is the broadly used antihistamine applied to treat allergic diseases. It has often been detected as an aqueous pollutant, indicating high stability in the environment [42]. Chemical oxidation methods were tested for cetirizine removal using sodium hypochlorite with a low degradation effect and UV-catalyzed oxidation resulting in high degradation efficiency [43,44].

Chlorhexidine and triclosan are broadly used disinfectants. They are detectable in high concentrations in hospital and domestic wastewaters [45]. Photocatalytic methods were tested for detoxification of chlorhexidine and triclosan [46,47,48]. In the case of triclosan, extremely toxic polychlorinated dibenzo-p-dioxines (PCDDs) were identified as oxidative by-products [48].

Fluconazole and ketoconazole are commonly used antifungal agents [49]. Photocatalytic processes were described as effective for the treatment of these antifungal drugs [50].

Hydrochlorothiazide is widely used diuretic [51]. The ecotoxicity of this drug and even of its oxidation products is mentioned in the literature [52]. The composition of starting aqueous solution contaminated with the above-mentioned nine halogenated biocides (WW) and the effect of tested reductants is depicted in Table 4. It is evident that the application of sole NaBH_4_ has a negligible effect on pollutants removal (Table 4, Run 1). This observation is in good agreement with the known fact that NaBH_4_ is not an effective HDH agent without the co-action of metal catalysts. By using the Raney Al-Ni/NaOH procedure, the concentrations of pollutants decrease significantly; however, the concentration of some of them (Triclosan, DCF, Fluconazole and Simazine) remains over 1 μg L^−1^ (Table 4, Run 2). Similarly, by applying the NaBH_4_/NiSO_4_ procedure (from NiSO_4_ obtained during recycling of spent Al-Ni alloy [38]), some of the pollutants are removed effectively; however, concentrations of triclosan, DCF, Atrazine and Simazine still remain over 1 μg L^−1^ (Table 4, Run 3).

By applying such an approach followed by the pretreatment by NiSO_4_/NaBH_4_ and subsequent Al-Ni/NaOH reduction, it is thus clearly possible to completely remove the majority of halogenated species from the model wastewater sample (i.e., the concentration of the respective halogenated pollutants decreases from ca 10–16 μg L^−1^ to max. 0.3 μg L^−1^; see Run 4 in Table 4).

### 3.5. Applicability of Ni-Based HDC for Treatment of the Real Water Matrix

For another demonstration of the robustness of the presented Al-Ni-based HDC concept, the efficiency of the Al-Ni alloy was evaluated using hospital wastewater spiked with DCF (1 mM DCF). Due to the stabilization of this hospital wastewater with added HNO_3_, the number of reducible compounds in the spiked hospital wastewater was high ([NO_3_^−^] = 1213.6 mg L^−1^; AOX = 73.5 mg Cl L^−1^; BOD_5_ = 219 mg O_2_ L^−1^; COD_Cr_ = 1374 mg O_2_ L^−1^, another characteristics are mentioned Section 2.1).

Based on these circumstances, the preliminary reduction in easily reducible compounds was performed by the addition of 1 M aqueous NiSO_4_ and followed by a dropwise addition of 12 wt.% NaBH_4_ in 14 M aqueous NaOH for alkalization of the treated water solution. In the subsequent step, Al-Ni alloy was added, and the resulting suspension was stirred for 240 min. Afterward, the obtained filtrate contained only APA according to the HPLC while AOX = 0.17 mg Cl L^−1^. After neutralization and separation of Al(OH)_3_, the biodegradability was determined by measurement of ratio BOD/COD = 0.485, which indicates easy biodegradability of organic compounds in the resulting treated wastewater. No significant inhibition effect on HDC, caused by high inorganic chloride content in this hospital wastewater, was observed in this case (see Table 5).

## 4. Conclusions

This study investigated the potential use of cheap and commercially available Al-Ni alloy within catalytic HDC of DCF in an alkaline aqueous solution. It was determined that commercial Raney Al-Ni alloy exhibited the highest HDC reactivity of DCF in the diluted aqueous KOH or NaOH solution. DCF, even in a concentration of 2 mM (0.59 g L^−1^ DCF), was completely dechlorinated into APA within 50 min by 1.1 g L^−1^ Al-Ni loading using 5.6 g L^−1^ KOH at room temperature. After the HDC process, the aqueous phase was neutralized and separated from the precipitated Al(OH)_3_. In the obtained filtrate, the remaining organic compounds are easily biodegradable. The recyclability of the used Al-Ni with or without the co-action of the aqueous solution of NaBH_4_ in NaOH was tested. Even under these conditions, HDC efficiency above 50% was achieved. In general, we demonstrated that a decrease in Al content in Al-Ni alloy is accompanied by a decrease in HDC activity of Al-Ni even when an additional reductant was added. When using NiSO_4_ with an excess of NaBH_4_ as the source of in situ generated HDC (i.e., Ni^0^) agent, only incomplete HDC efficiency of DCF was observed. On the other hand, we demonstrated that NaBH_4_/NiSO_4_ reagent is applicable for the pretreatment of a mixture of halogenated biocides in an aqueous solution or for the pretreatment of real hospital wastewater containing high concentrations of chlorides, reducible nitrates and DCF. Subsequent application of Al-Ni alloy to the obtained alkaline pretreated wastewater enables the completion of HDC of chlorinated aromatic or heterocyclic contaminants.

We proved that the concentration of nickel in treated aqueous solution after neutralization and separation of insoluble part is below 10 μg Ni/L due to the treatment caused by coagulation and flocculation of dissolved Al(III) species (precipitation of Al(OH)_3_), which acts as a sorbent of the Ni metal particles. Used Ni is completely recyclable (similarly to catalysts based on noble metals), as we documented in our previous works [33].

The described method was tested for real wastewater treatment with a content of chlorinated benzenes [53].

## Data Availability

Not applicable.

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
