# Peer review of "Application of Raney Al-Ni Alloy for Simple Hydrodehalogenation of Diclofenac and Other Halogenated Biocidal Contaminants in Alkaline Aqueous Solution under Ambient Conditions"

_materials, 2022, doi:10.3390/ma15113939_

Round 1

Reviewer 1 Report

Bendová reported the use of Al-Ni alloy to dehalogenate diclofenac in water systems. It is a good attempt. Unfortunately, the methods are confusing and the clarity of report is greatly obscured. 

  1. Extensive English language editing is required.
  2. A global picture is lacking in the introduction based on sources of data.
  3. Line 88. How can we ameliorate the toxicity of Ni? Its a problem so long as it is in the treated system.
  4. Line 117. Many details are lacking in the method section. What was the extraction method employed for DCF?
  5. Line 125. Did the authors use a uv detector (wavelength 283 nm)? If yes, then it conflicts with the information in line 121, "DAD detector."
  6. Line 113 and Table 1. Its confusing exactly how the authors carried out the batch experiments. Was there any form of repetition of measurements? 
  7. Line 123. How was the final concentration o DCF derived, taking into account of the sample dilution? 

Author Response

Responses of authors to the reviewer´s 1 comments:

Bendová reported the use of Al-Ni alloy to dehalogenate diclofenac in water systems. It is a good attempt. Unfortunately, the methods are confusing and the clarity of report is greatly obscured. 

  1. Extensive English language editing is required.

Answer: We apologize for our English language, we are not native English speakers. We hope that the new version is easier-to-understand.

  1. A global picture is lacking in the introduction based on sources of data.

Answer: We hope that „Introduction“ chapter is improved according to your comments.

  1. Line 88. How can we ameliorate the toxicity of Ni? Its a problem so long as it is in the treated system.

Answer: Using Al-Ni alloy, the metallic Ni and its alloys is produced. Most of them is simply separable by filtration. The residue is simply and very effectively removable using coagulation/flocculation of dissolved aluminium during neutralization step.

  1. Line 117. Many details are lacking in the method section. What was the extraction method employed for DCF?

Answer: The description is added in the actual version of our manuscript.

  1. Line 125. Did the authors use a uv detector (wavelength 283 nm)? If yes, then it conflicts with the information in line 121, "DAD detector."

Answer: The 1260 Infinity II DAD detector offers multiple wavelength and full spectral detection in a wide wavelength range of 190 to 950 nm due to dual-lamp design.

  1. Line 113 and Table 1. Its confusing exactly how the authors carried out the batch experiments. Was there any form of repetition of measurements? 

Answer: Some experiments were repeated and the agreement of the results was satisfactory (max. up to 5%).

  1. Line 123. How was the final concentration o DCF derived, taking into account of the sample dilution? 

Answer: The sample dilution was accurate (5 mL to 25 mL), so the determination of the final concentration of DFC was not problematic.

Reviewer 2 Report

 Abstract

Check grammar and add the missing punctuation marks according to English grammar (this observation is valid for the whole paper).

I recommend that the authors rephrase this paragraph to make it clear what the purpose of the experiment was, what materials they used (it should be specified at the beginning if real or synthetic hospital wastewater was tested, not at the end of the abstract. It was only in Materials and methods suggested that it is hospital wastewater, but again, is it real or synthetic?), which were the most important results (with values, preferably) and what is the practical utility of the results.

Use wastewater instead of waste water (in the entire paper)

Lines 26-27 are unclear, due to the use of “decontamination” – hospital wastewater contains a wide array of contaminants (physical, biological, chemical) – so you should be more specific and name the contaminants, use “the removal of …”.

Keywords

It is preferable that in this section not to repeat words that are also found in the title, but to find other defining words for the subject approached in the paper.

Introduction

English grammar is poor and needs to be improved.

Pay more attention to phrasing.

Indeed, emerging pharmaceutical pollutants are non-biodegradable and difficult to degrade in conventional treatment plants, but it should also be mentioned that some advanced treatment processes (eg AOPs, Fenton) have an increased efficiency in retaining them. AOPs are mentioned quite late (line 56).

Line 41 starts very abruptly and has nothing to do with the previous paragraph. It should be noted that the pollutants emerging from the effluents reach the receiving watercourses and that they have negative effects on the aquatic fauna and flora, at least.

Use italics for Oncorhynchus mykiss.

Line 47 – instead of “described effects cited” write “mentioned negative effects”.

Line 50 - instead of “animated nature including human population” write “all living organisms”.

Line 53 – “in the obtained aqueous concentrate” – be more specific, what type is it? Pay more attention when writing a statement. Did you obtained it yourselves or did the authors you quoted?

Line 54 – “destructive treatment methods” – it sound wrong (destructive for the contaminants, or for the aquatic environment? please change destructive and be more specific.

Lines 59, 60 – use papers instead of articles, and have you quoted those papers dealing with application of AOPs for DCF removal only quanti-60 fied DCF content in the treated aqueous solution?

Be more specific when using the term treated aqueous solution. Is it the same solution (of what) each time?

Avoid using formulations such as “in the relevant literature / most of the published articles / as it has been published already etc” because they sound bad in a scientific paper – just make the statement and then quote the source.

Line 74 – toxic to what or to who? Use complete sentences!

Rephrase lines 76-79 to state more clearly the aim of this study.

Materials and methods

Line 108 – this is wrong. “Experiments” involve people carrying them out, devices and other materials; hence they cannot be carried out in a flask. But they can be carried out in laboratory (in your case) or in field conditions.

Line 109 - the reaction, the reaction flask. Here, the flask is sufficient.

The method is described sequentially and in a manner that allows replication of experiments.

Results and discussion

The experimental results are compared with the results of similar studies.

I recommend editing the Figures to improve their quality.

Line 296 – the environment.

Conclusion

The conclusions are based on the description of the results obtained. It would be interesting to mention whether the method tested at laboratory level can be widely applied in a real wastewater treatment plant with content of emerging contaminants and possibly, if the method would be more feasible in terms of efficiency but also economically, compared to other existing methods for the removal of ECs.

Author Response

Reviewer comments and Suggestions for Authors:

 Abstract

Check grammar and add the missing punctuation marks according to English grammar (this observation is valid for the whole paper).

Answer: We apologize for our English language, we are not native English speakers. We hope that the new version is easier-to-understand.

I recommend that the authors rephrase this paragraph to make it clear what the purpose of the experiment was, what materials they used (it should be specified at the beginning if real or synthetic hospital wastewater was tested, not at the end of the abstract. It was only in Materials and methods suggested that it is hospital wastewater, but again, is it real or synthetic?), which were the most important results (with values, preferably) and what is the practical utility of the results.

Use wastewater instead of waste water (in the entire paper)

Answer: We used model aqueous solutions of described halogenated pollutants initially and even proved action of Al-Ni/OH- methodology for real hospital wastewater spiked with diclofenac, as we describe in the manuscript.

Lines 26-27 are unclear, due to the use of “decontamination” – hospital wastewater contains a wide array of contaminants (physical, biological, chemical) – so you should be more specific and name the contaminants, use “the removal of …”.

Answer: Thank you very much for your comment, it is clarified in the actual version of our manuscript.

Keywords

It is preferable that in this section not to repeat words that are also found in the title, but to find other defining words for the subject approached in the paper.

Answer: Thank you very much for your comment. We changed keywords which are not used in the title.

Introduction

English grammar is poor and needs to be improved.

Pay more attention to phrasing.

Answer: We apologize for our English language, we are not native English speakers. We hope that the new version is easier-to-understand.

Indeed, emerging pharmaceutical pollutants are non-biodegradable and difficult to degrade in conventional treatment plants, but it should also be mentioned that some advanced treatment processes (eg AOPs, Fenton) have an increased efficiency in retaining them. AOPs are mentioned quite late (line 56).

Answer: Thank you very much for your suggestion, we add newly published information dealing with AOPs into Introduction chapter.

Line 41 starts very abruptly and has nothing to do with the previous paragraph. It should be noted that the pollutants emerging from the effluents reach the receiving watercourses and that they have negative effects on the aquatic fauna and flora, at least.

Answer: Thank you very much for your suggestion, we add the above mentioned sentence.

Use italics for Oncorhynchus mykiss.

Answer: Thank you very much, the Italics style is used in the actual version.

Line 47 – instead of “described effects cited” write “mentioned negative effects”.

Line 50 - instead of “animated nature including human population” write “all living organisms”.

Line 53 – “in the obtained aqueous concentrate” – be more specific, what type is it? Pay more attention when writing a statement. Did you obtained it yourselves or did the authors you quoted?

Line 54 – “destructive treatment methods” – it sound wrong (destructive for the contaminants, or for the aquatic environment? please change destructive and be more specific.

Answer: Thank you very much, we utilized all the above mentioned suggestions in the actual version of our manuscript.

Lines 59, 60 – use papers instead of articles, and have you quoted those papers dealing with application of AOPs for DCF removal only quanti-60 fied DCF content in the treated aqueous solution?

Answer: The papers dealing with decrease of DCF concentration was added to the References chapter.

Be more specific when using the term treated aqueous solution. Is it the same solution (of what) each time?

Answer: We discuss the results published in References No. 16-19 or  in References No. 4,15,23,24.

Avoid using formulations such as “in the relevant literature / most of the published articles / as it has been published already etc” because they sound bad in a scientific paper – just make the statement and then quote the source.

Line 74 – toxic to what or to who? Use complete sentences!

Rephrase lines 76-79 to state more clearly the aim of this study.

 Answer: Thank you very much, we utilized all the above mentioned suggestions in the actual version of our manuscript.

Materials and methods

Line 108 – this is wrong. “Experiments” involve people carrying them out, devices and other materials; hence they cannot be carried out in a flask. But they can be carried out in laboratory (in your case) or in field conditions.

Line 109 - the reaction, the reaction flask. Here, the flask is sufficient.

 Answer: Thank you very much, we utilized all the above mentioned suggestions in the actual version of our manuscript.

The method is described sequentially and in a manner that allows replication of experiments.

Results and discussion

The experimental results are compared with the results of similar studies.

I recommend editing the Figures to improve their quality.

 Answer: Thank you very much for this comment, we edited the Figures in the actual version of our manuscript.

Line 296 – the environment.

Conclusion

The conclusions are based on the description of the results obtained. It would be interesting to mention whether the method tested at laboratory level can be widely applied in a real wastewater treatment plant with content of emerging contaminants and possibly, if the method would be more feasible in terms of efficiency but also economically, compared to other existing methods for the removal of ECs.

 Answer: Thank you very much for this comment, we add the information dealing with pilot-scale application of described HDH method.

Reviewer 3 Report

The manuscript materials-1708852 deals with an important topic - the removal of refractive pollutants from wastewater. The advantage is that partial studies (presented in the manuscript) are important for the broader technology of hospital wastewater treatment. Overall, the manuscript is of good quality.

The presented manuscript has some disadvantages as well.

  1. A lot of information is included in the additional materials. It seems like too much (equations of kinetic models, SEM results, table S3 and wastewater characteristics). There is no consistency: COD and BOD measure methods are in the main manuscript but the results are in SM, while the SEM method is described in SM and SEM results are in SM too. Overall, MS is where I expect additional information, but information that was originally already in the manuscript, e.g. subsequent research series that are already basically shown in the main document. However, the authors present a lot of qualitatively new results in SM (SEM, treatment of real wastewater), only mentioned in the main text. Because of this, the structure of the manuscript appears to be disrupted.
  2. The introduction lacks the characteristics of hospital wastewater and the problem of refractive pollution and the current concepts of solving this problem, also the relationship of the presented results with this topic.
  3. No coefficients or error functions were used to evaluate the fit of the kinetic models
  4. In paragraph 3.3 are presented results of Al-Ni alloy reuse. In the title is the "recycling" word, however, recycling involves recovering raw materials from waste products and using them to produce new goods. Please, rethink the applied term.
  5. The results from Section 3.4. seem to be a side topic. What is the relationship of the studied biocides with hospital wastewater? This text is less important for the entire article than, for example, SEM analysis.
  6. lines 336,337: please give the value of the BOD/COD ratio for used untreated wastewater.
  7. line 132 – LCK 555 is of BOD and LCK 515 of COD, reverse order of the methods numbers
  8. Table 3 is placed after references – a surprising position

Author Response

The presented manuscript has some disadvantages as well.

  1. A lot of information is included in the additional materials. It seems like too much (equations of kinetic models, SEM results, table S3 and wastewater characteristics). There is no consistency: COD and BOD measure methods are in the main manuscript but the results are in SM, while the SEM method is described in SM and SEM results are in SM too. Overall, MS is where I expect additional information, but information that was originally already in the manuscript, e.g. subsequent research series that are already basically shown in the main document. However, the authors present a lot of qualitatively new results in SM (SEM, treatment of real wastewater), only mentioned in the main text. Because of this, the structure of the manuscript appears to be disrupted.

Answer: Thank you very much for your suggestions. The actual version of our manuscript is significantly re-arranged. We hope that this version is more interesting for Materials readers.

  1. The introduction lacks the characteristics of hospital wastewater and the problem of refractive pollution and the current concepts of solving this problem, also the relationship of the presented results with this topic.

Answer: Thank you very much for this comment. The actual parameters of used hospital wastewater was added into the text of manuscript.

  1. No coefficients or error functions were used to evaluate the fit of the kinetic models

Answer: The values of kinetic constants k1 and k2 were obtained by minimizing of the standard deviation of experimental and calculated values of concentrations of all components of mixture (DCF, Cl-APA and APA). The standard deviation was in the limits from 6% to 20%.

  1. In paragraph 3.3 are presented results of Al-Ni alloy reuse. In the title is the "recycling" word, however, recycling involves recovering raw materials from waste products and using them to produce new goods. Please, rethink the applied term.

Answer: Thank you very much for this comment. The actual version of manuscript contains more correct term reused.

  1. The results from Section 3.4. seem to be a side topic. What is the relationship of the studied biocides with hospital wastewater? This text is less important for the entire article than, for example, SEM analysis.
  2. lines 336,337: please give the value of the BOD/COD ratio for used untreated wastewater.
  3. line 132 – LCK 555 is of BOD and LCK 515 of COD, reverse order of the methods numbers

Answer: Thank you very much for these suggestions. It is corrected in our actual version of manuscript.

  1. Table 3 is placed after references – a surprising position

Answer: We hope that this Table will be relocated by typesetters.

Reviewer 4 Report

The Manuscript “Application of Raney Al-Ni alloy for simple hydrodehalogenation of Diclofenac and other halogenated biocidal contaminants in alkaline aqueous solution under ambient conditions” requires revision before accepted for publication. The specific comments are given below.

  1. In 3 Analytical methods chapter, please indicate the manufacturer, city, country when mentioning the equipment.
  2. Statistical analyzes are very important in research manuscripts. How were the normality of the distribution and the differences between the variables found?
  3. Laboratory tests require repeatability. Add standard deviations to the results in tables, figures, and text. Write down the number of times the repetitions were performed.
  4. Present the results obtained by other authors in a tabular form.

Author Response

Reviewer comments and Suggestions for Authors:

The Manuscript “Application of Raney Al-Ni alloy for simple hydrodehalogenation of Diclofenac and other halogenated biocidal contaminants in alkaline aqueous solution under ambient conditions” requires revision before accepted for publication. The specific comments are given below.

  1. In 3 Analytical methods chapter, please indicate the manufacturer, city, country when mentioning the equipment.

Answer: Thank you very much for this comment. The indications are supplemented in the new version of our manuscript.

  1. Statistical analyzes are very important in research manuscripts. How were the normality of the distribution and the differences between the variables found?
  2. Laboratory tests require repeatability. Add standard deviations to the results in tables, figures, and text. Write down the number of times the repetitions were performed.

Answer: Thank you very much for your suggestions. We used equipment for the parallel reaction performance (Starfish attachment) for kinetic studies. The experiments were performed once using this technique, however, the time behaviour of sampling from different reaction flasks overlapped. The values of kinetic constants k1 and k2 were obtained by minimizing of the standard deviation of experimental and calculated values of concentrations of all components of mixture (DCF, Cl-APA and APA). The standard deviation was in the limits from 6% to 20%.

  1. Present the results obtained by other authors in a tabular form.

Answer: Thank you very much for your suggestion. The mentioned table is added into the actual version of our manuscript.

Round 2

Reviewer 1 Report

The authors seem to have addressed my concerns. 

Reviewer 4 Report

Thank you for considering my suggestions